# Rheumatoid Meningitis a Rare Extra-Articular Manifestation of Rheumatoid Arthritis: Report of 6 Cases and Literature Review

**DOI:** 10.3390/jcm9061625

**Published:** 2020-05-27

**Authors:** Mélanie Trabelsi, Xavier Romand, Mélanie Gilson, Mathieu Vaillant, Pierre-André Guerne, Gilles Hayem, Ewa Bertolini, Athan Baillet, Philippe Gaudin

**Affiliations:** 1Rheumatology Department, CHU Grenoble Alpes, Hôpital Sud, 38130 Echirolles, France; xromand@chu-grenoble.fr (X.R.); mgilson@chu-grenoble.fr (M.G.); abaillet@chu-grenoble.fr (A.B.); pgaudin@chu-grenoble.fr (P.G.); 2GREPI TIMC, CNRS UMR 5525, Université Grenoble Alpes, 38000 Grenoble, France; 3Neurology Department, CHU Grenoble Alpes, Hôpital Nord, 38000 Grenoble, France; mvaillant@chu-grenoble.fr; 4Rheumatology Department, CHU Genève, 1205 Geneva, Switzerland; drpaguerne@bluewin.ch; 5Rheumatology Department, Groupe Hospitalier Paris Saint Joseph, 75014 Paris, France; ghayem@hpsj.fr; 6Rheumatology Department, CH Annecy, 74370 Annecy, France; ebertolini@ch-annecygenevois.fr

**Keywords:** rheumatoid arthritis, aseptic meningitis, leptomeningitis, pachymeningitis

## Abstract

Objectives. Central neurological manifestations of rheumatoid arthritis (RA) like rheumatoid meningitis (RM) are rare, little known and have a high rate of morbi-mortality. METHODS. We described six cases of RM that were directly related to RA activity after exhaustive assessment. RESULTS. They were mainly women, aged of 50 to 69. All were positive for anti-cyclic citrullinated peptide antibodies and half for rheumatoid factors. RA activity, duration, and treatments were heterogeneous including oral steroids, conventional synthetic disease modifying anti-rheumatic drugs (DMARDs) and biologic DMARDs. Symptoms were various, with acute or progressive beginning; main were: generalized or focal seizure (4/6), fever (3/6), headaches (3/6), and frontal syndrome (2/6). Imaging lesions were four leptomeningitis, one pachymeningitis, and one association of both. MRI usually showed hypersignal in various territories in T2-FLAIR (fluid attenuated inversion recovery) mode, and enhancement in T1-weighted mode after gadolinium injection. All patients had lumbar puncture that found sterile cerebrospinal fluid, no neoplasic cell, elevated cell count in 5/6 cases and elevated proteins concentration in 3/6 cases. Cerebral biopsy was possible for three patients, and definitively confirmed the diagnosis of aseptic lepto- or pachymenintis, excluding vasculitis and lymphoma. Different treatments were used like intravenous high dose steroids, immunoglobulins or biologic DMARDs, with variable clinical and imaging outcome: one death, one complete recovery, and four recoveries with sequelae. Conclusions. Clinical symptoms, imaging, lumbar puncture, and serological studies are often nonspecific, only histologic examination can confirm the diagnosis of RM. Any central neurological manifestation in RA patients, even in quiescent and ancient RA, should warn the physician.

## 1. Introduction

Rheumatoid arthritis (RA) is the most common chronic inflammatory arthritis, leading to joint damage, degradation of quality of life, disability, and decrease of life expectancy. 

Extra articular manifestations are described in about 20% of patients. Sites of extra-articular manifestations include skin, eyes, lungs, kidneys, heart, and blood vessels. These complications represent bad prognosis criteria because they often concern vital prognosis. Among extra-articular manifestations, neurological complications are rare [1], and peripheral neurological symptoms are the most common. Brain involvement in RA includes vasculitis, leading to hemorrhage or strokes, rheumatoid lepto- or pachy-meningitis, and rheumatoid nodule formation, whereas normal-pressure hydrocephalus has also been reported [2,3,4,5]. Rheumatoid meningitis (RM) including lepto- and pachy-meningitis is an aseptic inflammation of meninges which is a rare and severe manifestation of central nervous system involvement (CNS) in RA [6]. 

The first case was described in 1949. Then, the largest case series described 19 patients in 1989 [6]. In most cases, onset occurred after the age of 50 in both sexes. Although most often seen in patients with longstanding seropositive RA with extra-articular involvement, it may even present in patients with no prior history of RA, with RM as the presenting manifestation, and occurs independently of disease activity or RA treatments [7,8]. In an autopsy series, there was poor correlation between the severity of systemic RA and neurologic manifestations [6,9]. 

Diagnosis is a major issue because RM can present as variable presentation: cognitive dysfunction, sensory or motor dysfunction, altered mental status, seizure, headache, or focal neurologic signs, depending on the brain structures involved. All of these can mimic either stroke, intracranial tumor, Parkinson syndromes, or even progressive supranuclear palsy [10,11,12,13,14]. Antemortem RM diagnosis is often difficult, several cases report its diagnosis at the time of autopsy.

Imaging and serological studies are variable, often nonspecific, and there is currently no serological pattern of RM. Serum levels of rheumatoid factors (RF) and anti-cyclic citrullinated peptide antibodies (ACPA) often only confirm the diagnosis of RA. Exact pathology remains unclear but it is associated with infiltration of mononuclear cells around small vessels in the leptomeninges. Cerebrospinal fluid (CSF) analysis tends to show an increase in inflammatory cells (mostly mononuclear), an elevation in protein, but lumbar puncture can also be normal [6,7]. Because of better access to magnetic resonance imaging (MRI), RM has been increasingly diagnosed. MRI lesions most described are pachy- and/or lepto-meningeal thickening with contrast enhancement, but they are not firmly specific of RM [15]. In the end, definitive diagnosis is based on histologic slides which are usually performed to confirm the etiology of these uncommon neurological manifestations. 

A high rate of morbidity was historically observed in RM, and mortality was above 60% even with treatment [1]. Therefore, high doses of steroids may not be sufficient and other treatments may be needed. But no standardized treatment is actually established. 

Here, we report six recent cases of RM by describing clinical, MRI, biological, and histological patterns, treatment, and outcome in order to better assess RM pattern. 

## 2. Methods

### 2.1. Study Design

This was a retrospective study performed in the Rheumatology and Neurology departments of the University hospital of Grenoble, France. 

After the occurrence of three recent cases of central neurological manifestations certainly related to RA among our cohort of RA patients, we decided since 2010 to collect more cases in order to better describe this rare and little known complication of RA (pattern of RA patients, natural history and symptoms, histological and MRI patterns, treatment and outcome). Thus, we contacted our colleagues specialized in neurology, histopathology, and radiology in our university center. Under the auspices of the Club Rhumatismes et Inflammations (CRI), a section of the French Society of Rheumatology and a partnership with the French Society of Internal Medicine, 1800 French and French speaking rheumatologists were informed through their website (www.cri-net.com) about this retrospective study. 

### 2.2. Definition of Cases 

Cases were patients having RA according to the American College of Rheumatology (ACR) 2010 criteria, and having developed any central neurological disorder which was directly related to RA activity after exhaustive assessment. Other diagnosis of central neurological manifestations such as atlanto-axial subluxation, infectious and malignant processes, and adverse events related to RA treatments were excluded from this study. 

#### 2.2.1. Data Collection 

Data concerning demography, RA duration, presence of RF, ACPA, and joint erosions on radiological exams were retrospectively collected. RA activity at the time of first neurological symptoms was assessed by the 28-joints Disease Activity Score (DAS28) and was classified as remission (<2.6), low (2.6–3.2), moderate (3.2–5.1), and high (>5.1) activity. As far as neurological complications of RA are concerned, symptoms and their way of occurrence, time from first neurological symptoms, and diagnosis of central complication of RA, MRI, and histological patterns were systematically collected. Finally, treatment and outcome at the time of last news was reported. 

#### 2.2.2. Statistical Analysis 

Descriptive analyses of the data were performed.

#### 2.2.3. Ethical Approval Information

This study was conducted in compliance with the protocol of Good Clinical Practices. In accordance with French law, formal approval from an ethical committee was not required for this observational retrospective study.

#### 2.2.4. Literature Research

Literature data were extracted from PubMed database in order to approximately evaluate the frequency of this complication of RA, and to compare our cases to cases from other centers as far as pattern of RA patients, natural history, symptoms, histological and MRI patterns, treatment and outcome are concerned. MeSH (Medical Subjects Headings) vocabulary keywords searched were: “arthritis, rheumatoid AND meningitis”; “arthritis, rheumatoid AND meningitis, aseptic”; “arthritis, rheumatoid/drug therapy AND meningitis”; “arthritis, rheumatoid/complications AND meningitis”. Only cases including exhaustive data ensuring the diagnosis of central neurological manifestations of RA were taken into account. 

## 3. Results 

### 3.1. Study Population 

A total of six cases of RM were analyzed from 2010 to 2017. Five from French Rheumatology centers (three from Grenoble university hospital, one from Paris university hospital, one from Annecy hospital) and one from Switzerland (Geneva university hospital). 

Main characteristics of cases (demography, RA profile, neurological symptoms, treatment and outcome of the central neurological complication of RA) are summarized in Table 1. 

They were mainly women (4 out of 6), with an average age of onset of the neurological involvement of 60.3 ± 5.9 years (average ± SD). Two patients had no extra articular impairment, one presented with pericarditis, one with pleuresia, one with subcutaneous nodules, and one with episcleritis. None had Sjogren’s syndrome associated. ACPA were positive for all and RF were positive for half. Two RA presented with erosions. One patient had no treatment, three were treated with an association of oral steroids and conventional synthetic disease-modifying antirheumatic drugs (csDMARDs) (including two under Methotrexate, and one under Hydroxychloroquine), two were treated with an association of Methotrexate and bDMARDs (Adalimumab). 

RM was diagnosed between 50 and 69 years old, with an average RA duration of 7.0 ± 8.4 years (average ± SD), going from 6 months to 25 years. The diagnosis of RM was established with average of 4.3 ± 2.5 months (average ± SD). At that time, one patient had low activity disease (tender joint count (TJC) = 0, swollen joint count (SJC) = 0), two had moderate activity (respectively TJC = 0 and 8, SJC = 9 and 2), one had high activity (TJC = « many », SJC = « many »), one was in remission (TJC = 0, SJC = 0), and one in flare. The accurate DAS28 (28-joints disease activity score) was not exactly calculable for three patients, because of missing data, nevertheless all were classified in different categories of disease activity.

### 3.2. Central Neurological Symptoms 

The symptoms beginning was equally progressive or acute. Symptoms observed were mainly generalized or focal seizure (4/6), fever (3/6), headaches (3/6), and frontal syndrome (2/6). We also observed abnormal movements of the lower limb (1/6), alteration of the general state (1/6), coma (1/6), delirium (1/6), dizziness with loss of consciousness (1/6), psycho motor retardation (1/6), depression anxiety syndrome (1/6). 

### 3.3. Cerebral Imaging

Type of cerebral imaging lesions were mainly leptomeningitis (4 out of 6 patients), but also one pachymeningitis and one association lepto and pachymeningitis. No intra parenchymal lesion was observed. 

MRI data found diffuse lesions, concerning frontal, parietal and/or temporal territories. MRI showed a meninges thickening with hypersignal in T2-weighted images, in T2-weighted-FLAIR (fluid-attenuated inversion recovery) mode and enhancement in T1-weighted images after intravenous (IV) gadolinium injection (four patients, data were missing for the two others) (Figure 1). 

### 3.4. Lumbar Puncture 

All patients underwent lumbar puncture. CSF analysis was negative for bacteriological and neoplasic cell. CSF elevated cell count (ranging from 19 to 150 cells/mm^3^) was found in five cases and CSF elevated proteins concentration (ranging from 0.44 to 0.94 g/L) in three cases. Glycorrhachia was normal in four cases (data were missing for two others). CSF was strictly normal for one patient. 

### 3.5. Differential Diagnosis 

All patients underwent exhaustive assessment permitting to exclude differential diagnosis, as vasculitis, infectious disease, or tumors. 

Antinuclear antibodies, extractable nuclear antigens, antineutrophil cytoplasmic antibodies (ANCA), anti-myeloperoxidase, and anti-proteinase 3 were measured in most patients and were negative (except for one patient who had positive ANCA). Complement levels were measured in half patients (3/6) with 2 arguing for inflammatory syndrome. Type II cryoglobulinemia was found in one out of two patients. Only one patient underwent a salivary gland biopsy which showed a non-specific sialadenitis. Two patients performed an angio-imaging permitting to exclude the vasculitis diagnosis. Two patients got a normal checking with an ophthalmologist. 

Infectious causes of meningitis were ruled out. Three QuantiFERON TB tests were done with a negative result. Five patients had negative serologic testing for syphilis (Treponema Pallidum Hemagglutinations Assay (TPHA) and Venereal Disease Research Laboratory (VDRL)). Human 

Immunodeficiency virus (HIV), hepatitis B virus, hepatitis C virus, Epstein-Barr virus, herpes simplex virus (HSV), Varicella-Zoster virus (VZV), cytomegalovirus, toxoplasmosis or Lyme disease serologies were performed in six patients. All these serologies were negative. 

CSF data showed an absence of bacteriological or neoplasic cell. Two *Mycobacterium tuberculosis* polymerase chain reaction (PCR) assays were done on the CSF, one HIV PCR assay, and one Whipple PCR assay. All were negative. 

Two patients had lymphocytic immunophenotyping which were normal. Thoracic-abdominal-pelvic computerized tomogram (CT) scan were performed in five patients, all were normal. 

### 3.6. Cerebral Biopsy 

Brain MRI and CSF analysis are useful to guide the diagnosis, but not sufficient: cerebral biopsy, when possible, is necessary to firmly conclude. Only three patients underwent cerebral biopsy (the others were contra-indicated because of the location of the lesion and the risk of hemorrhage). All histopathological results found an absence of vasculitis, and no rheumatoid nodule. One found an inflammatory necrotic chronic leptomeningitis (meninges infiltration with T lymphocytes, macrophages, few B lymphocytes with positive CD-20 marking) noticeable in Figure 2. One found an inflammatory chronic pachymeningitis (meninge infiltration with T lymphocytes, B lymphocytes, plasmocytes, histiocytes). The other found a leptomeningitis with necrotic granuloma. 

### 3.7. Treatments 

Initially, five patients were treated with IV high dose steroids with heterogeneous dosage going from 250 mg to 1000 mg a day during 3 to 10 days. They continued oral moderate dose of steroids after IV pulses. This was successful for three patients and led to clinical remission of neurological symptoms. One patient did not receive IV steroids nor immunosuppressive treatment but continued his regular oral moderate dose of steroids with a neurological clinical improvement. 

Then, different treatments were used in order to decrease or stop oral steroids and to reduce RA activity. One received Rituximab with clinical improvement. One received Etanercept which was not efficient on neurological symptoms and then Rituximab but he passed away. One received Etanercept then Adalimumab which were both not efficient on neurological symptoms. One patient received IV immunoglobulins (Ig) with clinical improvement. Two patients did not receive immunosuppressive therapies. 

Patients with seizure symptoms were treated with anticonvulsant drugs with good efficacy. 

Before the definitive diagnosis of RM, when an infectious cause was suspected, patients received empirical antibiotics and/or antiviral agents, without any efficacy: one received only Aciclovir, one received an association of Aciclovir, Gentamicine, and Amoxicilline, and one received Aciclovir then an anti-tuberculosis therapy. 

### 3.8. Outcome and Follow-Up 

The patients follow-up was approximately 2 years after the RM diagnosis, ranging from 11 months to 10 years. The neurological recovery of these patients varied: one complete recovery, four neurological sequelae, and one death by subarachnoid hemorrhage. Among these four patients, three presented stable chronic MRI lesions after several years of follow-up (long-term MRI data were missing for one patient).

## 4. Discussion 

This six case series emphasized that RM can present as variable clinical symptoms, and diagnosis can be confirmed only after exhaustive assessment and cerebral biopsy. RA neurological extra-articular symptoms are rare. The most frequent are rheumatoid vasculitis whose symptoms are skin ulcers, skin purpura, and mononeuropathy multiplex. Central neurological symptoms are also rare. They are mainly consequences of spine compression because of cervical vertebrae subluxation; cerebral hemorrhage linked to hyperviscosity because of RF in the serum; cerebral vasculitis; rheumatoid nodules; or inflammatory infiltrations of meninges: leptomeningitis and pachymeningitis. In the literature, less than 100 cases of aseptic rheumatoid lepto- or pachymeningitis are described, and the largest series (19 cases) is a historical one, before availability of biologic treatments. Cases of rheumatoid encephalitis are also described but are out of the scope of this study.

Even though RM is a rare condition, more and more case reports are described in the recent literature, and its mortality may increase by 70% as described in a 2003 review [1]. Like in our study, most of them had long-standing seropositive RA with erosions. But sometimes, neurological symptoms could happen before RA diagnosis [16] (more than 20% of cases). The most described clinical symptom in scientific literature is an altered mental status. Other symptoms at presentation were hemi/paraparesis or focal symptoms (seen in more than 60% of recent cases), headache (40% of cases), cranial nerve symptoms, or seizure which is the most common clinical symptom seen in our study. Acute neurologic deficit is a frequent complaint encountered by clinicians, who need to be aware of stroke mimickers. Clinical features may also mimic Parkinson syndromes or progressive supranuclear palsy. All these clinical presentations are summarized in Table 2, which presents a literature review since 2014 to 2019 of authentic RM cases.

### 4.1. Differential Diagnosis

Differential diagnosis may mainly concern infectious disease, vasculitis, or tumors. In our series, various tests were used to rule out all these differential diagnoses. Nevertheless, they were not homogeneous between the six patients and all did not performed exactly the same analysis or imaging. 

### 4.2. Infectious Diseases

CNS infections reported in patients with RA include progressive multifocal encephalopathy, aspergilloma, tuberculosis, syphilis, infection with West Nile virus, bacterial meningitis, viral meningitis (HSV, VZV, Coxsackie, measle, rubella), infection with rhodococcus and cryptococcal meningitis. Some meningitis due to Whipple’s disease are also described [17]. Because of its atypical presentation, it may be easily misdiagnosed. Patients who have been reported with opportunistic infections were mostly undergoing immune therapy [18]. This is why they usually received large empirical antibiotics or antiviral agents before the certain diagnosis is confirmed. In our series, all of these infectious causes were ruled out by performing multiple serologies, PCR assay, and culture (blood, CSF, brain biopsy). 

### 4.3. Vasculitis 

Lepto- or pachymeningitis can be caused by RA itself but also by primary CNS vasculitis or others systemic diseases like lupus, Sjogren syndrome, adult onset Still’s disease, sarcoidosis or Behçet disease that have to be ruled out with biological analysis, angio-imaging like computerized tomography (CT)angiogram, and sometimes by histological analysis. 

### 4.4. Tumoral Diseases 

Tumor has to be excluded with body CTscan and/or positron emission tomography (PET) scan. It is also important to complete this assessment with a lymphocytic immunophenotyping because aseptic meningitis can occur as a complication of lymphoma (especially malignant B-cell non-Hodgkin lymphoma) [19]. 

### 4.5. Drug-Induced Adverse Events 

Drug-induced aseptic meningitis may also be considered as a possible diagnosis. Several case reports have described aseptic meningitis that has resulted from medication use like anti-tumor necrosis factor alpha (anti-TNF α) agents as Infliximab or Adalimumab [20] and even from csDMARDs like Methotrexate, Salazopyrine, or Leflunomide [21,22,23]. Nevertheless, anti-TNF α therapy is most often associated with CNS events including new onset or exacerbations of pre-existing demyelinating neurological diseases. 

In the literature, more than 100 cases of drug-induced aseptic meningitis are described with various type of medication. Nonsteroidal anti-inflammatory drugs (NSAIDs), antibiotics, antituberculous agents or polyvalent human Ig may also be implicated in neurological attempt. 

#### 4.5.1. MRI 

MRI feature is crucial because it points out the meningeal origin of the symptoms and may support the diagnosis. But even if lesions are characteristic, they are not specific. Most MRI data concern hypersignal in T2 weighted mode, T2-weighted-FLAIR mode, and enhancement on gadolinium-enhanced T1 weighted imaging [24,25]. One of the radiological features of this disease is a tendency to present with unilateral supratentorial lesions [9,26,27]. 

#### 4.5.2. Lumbar Puncture 

Lumbar puncture usually shows abnormal undefined results: CSF increased cell count consistent with lymphocytic meningitis, mild increase of the CSF protein concentration, and normal glucose level [28]. Low glycorrhachia may suggest bacterial meningitis (especially *Mycobacterium tuberculosis*). Findings for infection (especially *Mycobacterium tuberculosis* PCR assay) or malignancy should always be checked. RF in the CSF seems to be specifically associated with meningeal disease and could be used as a biomarker [1,3]. A recent study suggests the role of anti-agalactosyl IgG antibody in the CSF as a helpful biomarker in diagnosis and assessment of the severity of RM [29]. Furthermore, we can wonder about the relevance of repeating examinations (laboratory, lumbar puncture, MRI) in the diagnosis approach. 

#### 4.5.3. Cerebral Biopsy 

The diagnosis of RM should be confirmed by biopsy. It can rule out others chronic meningitis such as fungal or mycobacterial infection, carcinomatosis, or lymphoma. There is no difference in histopathological findings between pachymeningitis and leptomeningitis [6,9], and the reason for the specific selection of pachymeninge or leptomeninge by invading inflammatory cells is unclear. Usually histologically finding is a lepto- and/or pachy-meninge thickening with lymphocytic aggregates (T and B lymphocytes, plasmocytes). Sometimes rheumatoid nodules or non-specific granulomatous lesions may be seen [30]. It can also show lymphocytic vasculitis of the cortical tissue and patchy lymphoplasmacytic infiltrates of dural small vessels [31]. Although meningeal infiltration is often noted, the presence of all typical findings in a single patient is rare. 

#### 4.5.4. RM Pathogenesis Hypothesis

Pathogenesis of RM is unknown and only some case reports studied possible mechanisms involved in this disease. Nissen et al. described an elevated number of B-lymphocytes (7.8%), plasma cells (1.8%) associated with a remarkable high ACPA level (19,000 UI/mL), high IgG index (1.45), and overexpression of CXCL13 in CSF of RA patient with RM. After IV high dose methylprednisolone and Rituximab treatment, ACPA and CXCL13 levels were seriously decreased in CSF in accordance with clinical symptoms resolution [32]. These data can suggest a local CNS synthesis of ACPA. CXCL13, a B-cell chemoattractant, drives the organization of the meningeal tertiary lymphoid organs, as described in RA synovitis [33] and correlates with intrathecal IgG production in antibody-mediated CNS disorder (neuromyelitis optica) and multiple sclerosis [34,35]. In RA synovitis are observed ectopic lymphoid structures with CD138+-plasma cells secreting ACPA [33]. ACPA are known to induce osteoclast differentiation, subsequent changes in nociception through IL-8 production and pro-inflammatory cytokines release by macrophages in joints [36]. We notice that all patients included in our study had ACPA positivity, in accordance with previous studies (Table 2). ACPA-positive patients are known to have more aggressive RA with a higher risk of subsequent radiographic joint damages and pulmonary extra-articular manifestations [37]. It has long been established that the brain was an immune-privileged organ. Recently, demonstration of patrolling leucocytes and the existence of lymphatic vessels in the meninges connected with the deep cervical lymph nodes have shifted this paradigm [38]. Hence, we assume that CXCL13 produce by meningeal stromal cells could attract B-cells secreting ACPA in meningeal tertiary lymphoid organs. Moreover, targeting B-cells with Rituximab was an effective treatment strategy in one patient in this study. Local production of plasmocytes from pre-B cells could drive localized immune RM process through ACPA production, irrespective of RA activity. Future researches are necessary to confirm this assumption.
jcm-09-01625-t002_Table 2Table 2Literature review since 2014 to 2019: cases of RM.Study *(ref)*Gender (F/M)Age of DiagnosisDisease HistoryAuto-AntibodiesNeurological SymptomsTreatmentOutcome (Follow-Up Duration)**Total of 15 studies**67% F63.7 years(+/−10.4) (mean(+/−SD))
>53% longstanding RA (≥9 years)>33% low activity>33% under treatment>26% erosive RA>26% diagnosis made during RM71% RF+ (5/7)* 100% ACPA+ (7/7)*67% hemi paraparesis or focal symptoms53% altered mental status40% headache27% seizure>86% steroids20% Cyclo-phosphamide27% clinical and/or MRI recovery27% partial recovery/sequelae20% unknown**1** [39]F50Several-years history of RA, csDMARD, low activityRF N/A, ACPA+Aphasia, confusion, transient leg weakness, headache, facial droopSteroidsRecovery (6 months)**2** [25]F84Diagnosis made during RMRF N/A, ACPA+Muscles weakness, cognitive dysfunctionIV steroid pulsesClinical and MRI improvement (?)**3** [7]M70Longstanding erosive RA, high activityRF+, ACPA N/AHemiparesis, sensitive symptoms, headacheIV steroid pulses, then oral steroids with Hydroxy chloroquine and SulfasalazineClinical recovery, MRI lesions improvement (5 months)**4** [12]M6810-years history of erosive RA, low activityRF N/A, ACPA N/AResting tremor, cognitive dysfunction, fever, bedriddenIV steroid pulsesRecovery with sequelae: parkinsonism and MRI hydrocephalus**5** [29]F69No arthritis, diagnosis made during RMRF N/A, ACPA+Aphasia, apraxia, hemiparesis, seizureIV steroid pulsesClinical and MRI improvement (?)**6** [40]F6023-years history of erosive RA, Auranofin and prednisoneRF+, ACPA+Headache, photophobia, delirium, impaired short-term memory, hallucinations, seizureIV steroid pulsesMRI improvement, clinical recovery with sequelae: minimal memory impairment**7** [41]M37Diagnosis made during RMRF+, ACPA+Headache, transient focal neurologic deficits, transient cognitive dysfunctionDexamethasone, Adalimumab + LeflunomidePartial improvement (?)**8** [41]F62Diagnosis made during RMRF-, ACPA N/APersonality change, lower limb weakness, confusion, seizures, tetraplegiaIV steroid pulses Recovery with sequelae: confusion, hallucination, paraparesia (few months)**9** [24]F6518-years history of RA, low activityRF N/A, ACPA N/ASeizure, hemiparesisIV steroid pulsesMRI and clinical improvement (?)**10** [28]M60Several-years history of RA, low activityRF N/A ACPA N/AHemiparesis, hypoesthesia, fever??**11** [31]F75RA with no treatmentRF+, ACPA N/ABehavioral changes, cognitive dysfunctionIV steroid pulses, CyclophosphamideMild improvement, then death (few months)**12** [42]F639-years history of RARF N/A, ACPA N/AHeadache, paraparesisSteroids, Cyclophosphamide?**13** [43]M5911-years history of on erosive RA, remission, csDMARDRF-, ACPA+Transient hemiparesis, hypoesthesia, feverOral steroids, RituximabClinical and MRI recovery (5 years)**14** [44]F716-years history of RA, low activity, csDMARD, anti-TNF alphaRF+, ACPA+Dysarthria, hemiparesthesia, headacheIV steroid pulses, CyclophosphamideClinical recovery, stable MRI lesions (6 months)**15** [45]F63? RA, anti-TNF alpha, steroidsRF N/A, ACPA N/AApathy, drowsiness??M: male, F: female, RA: rheumatoid arthritis, RM: rheumatoid meningitis, RF: rheumatoid factors, ACPA: anti citrullinated peptide antibodies, IV: intra venous, csDMARD: conventional synthetic disease modifying anti-rheumatic drug, TNF: tumor necrosis factor, MRI: magnetic resonance imaging, N/A: not available, * percentage of RA patient with auto-antibodies status available.


#### 4.5.5. Treatments

There is currently no guideline concerning RM treatment. Immunosuppressive treatment still remains the major therapeutic choice for systemic autoimmune disorders aiming to reduce systemic inflammation and prevent permanent damage caused by the disease. Corticosteroids remain the mainstay of immunotherapy, either in high oral or IV doses, while lower oral doses are routinely used for maintenance treatment. Other conventional immunomodulating drugs, including Methotrexate, Azathioprine, Cyclophosphamide, Mycophenolate mofetil, Ciclosporin, IV Ig and plasmapheresis have been tried [8,9,15,28,29]. Some studies pointed out the role of Cyclophosphamide in association with IV steroids pulses [26]. Anti-TNF α agents [46], monoclonal antibodies targeting B cells (Rituximab), inflammatory cytokines such as interleukins (IL)-6 and IL-1β, or co-stimulatory molecules (Abatacept) have been punctually tried as RM treatment. In 2009, Schmid et al. reported a case report, with efficiency of Rituximab in RM [5]. In our cases, one patient has been successfully treated by Rituximab but the other one died too quickly to consider any efficiency. Indeed, our three brain biopsies showed meninges infiltration with B lymphocytes confirmed with immunohistochemistry positive CD-20 marking. This treatment may be proposed because brain biopsy has shown a large amount of CD-20 expressing B lymphocytes.

Few published cases describe the occurrence of RM during anti-TNF α treatments, suspecting a potential contribution of anti-TNF α in it [10]. In any case, anti-TNF α therapy seems not to prevent RM beginning [47]. Further, Methotrexate has been reported to cause rheumatoid nodules on the meninges. Whatever, the efficacy of these agents has been demonstrated in small non-controlled clinical trials or case reports. They concern many systemic disorders (not only RA), and all neurological manifestations (peripheral and central). Finally, treatment for RM still remains empirical. 

#### 4.5.6. Outcome

Few data concerning prognosis are available, with a short follow-up duration (or often unknown). Clinical and imaging outcomes are variable. Some patients may rapidly improve, others may keep sequelae, or worsen [25,47]. But death occurred in many case series, especially before the emergence of immunosuppressive treatments.

There is currently no consensus about the treatment, and no data about the role of an early diagnosis with prompt aggressive treatment [8,15]. Nevertheless, therapeutic option should be considered as soon as possible because of the poor prognosis. We can wonder about the relevance of follow-up examinations (especially lumbar puncture and MRI). Indeed, several cases described CSF or imaging improvement while clinical improvement.

## 5. Conclusions

Clinical symptoms, imaging, lumbar puncture, and serological studies are often nonspecific. We could point out that females with a long-standing seropositive RA are more likely to develop RM. But only histologic examination can confirm the diagnosis. In these RA patients undergoing immunosuppressive treatments, it is important to exclude vasculitis, infectious, or tumorous causes. Differential diagnoses are profuse. It is necessary to exclude them with an exhaustive assessment and multidisciplinary advice, including at least: brain MRI, body CTscan, lumbar puncture with PCR assays (HSV, Mycobacterium tuberculosis, Whipple, Lyme), serological studies (quantiFERON TB test, HIV, hepatitis, TPHA-VDRL, EBV, CMV, VZV, and HSV tests, complete auto immune assessment, lymphocytic immunophenotyping). It is worth noting that there is currently no guideline of diagnostic approach and no standardized RM treatment.

In the end, any central neurological or psychiatric manifestation in RA patients, even in quiescent and ancient RA, should warn the physician. 

## Figures and Tables

**Figure 1 jcm-09-01625-f001:**
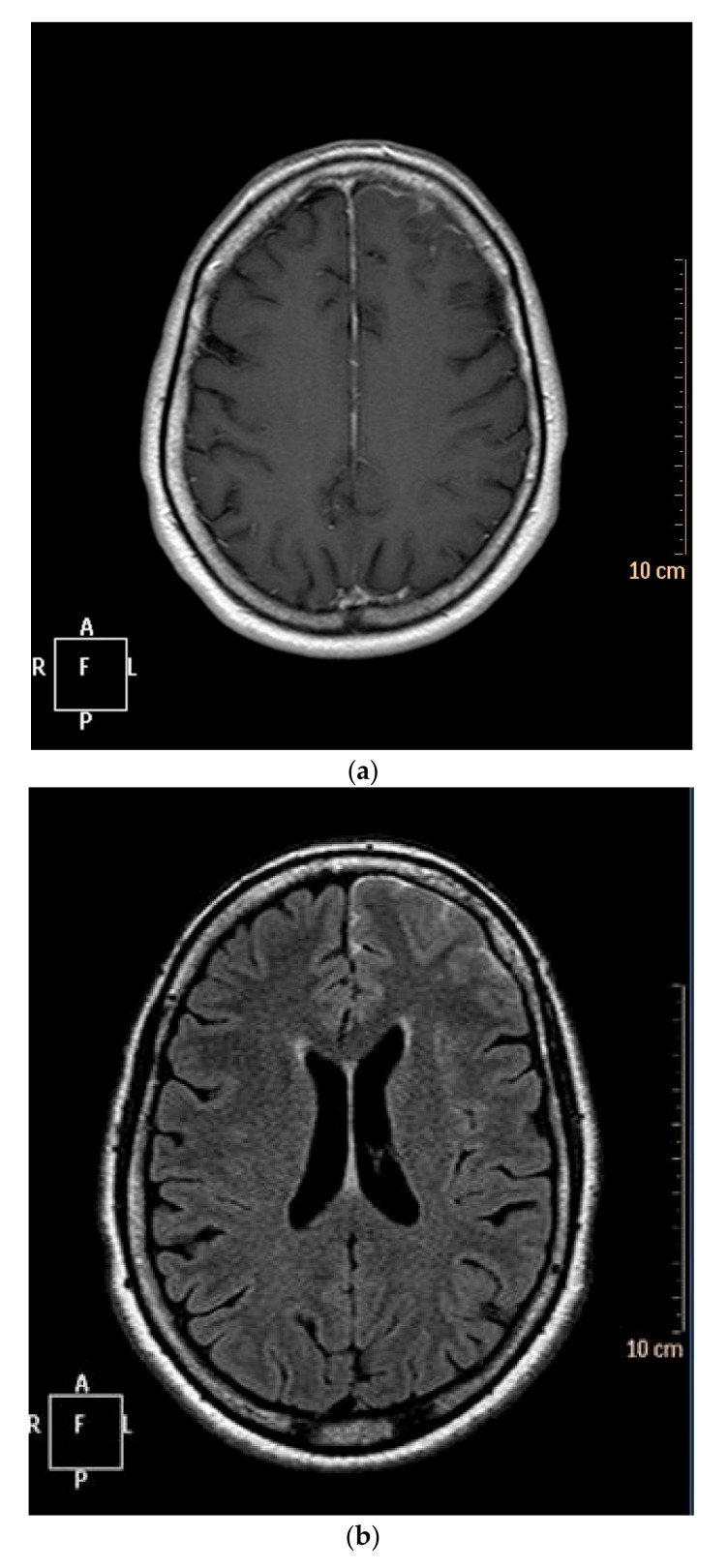
Cerebral MRI of our first case. (**a**) T1-weighted images after gadolinium injection, showing enhancement of the leptomeninx in the left frontal lobe. (**b**) T2-weighted-FLAIR images showing hypersignal in left frontal lobe.

**Figure 2 jcm-09-01625-f002:**
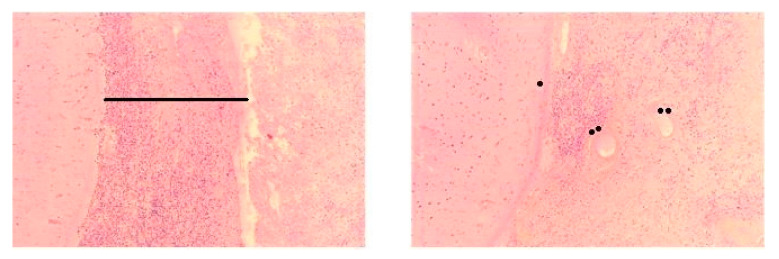
Cerebral biopsy of our second case. Coloration with hematoxylin eosin safran. Light magnification. Leptomeninx thickening with inflammatory infiltration composed by lymphoplasmocytes (black line), vascular wall hyaline fiber (one point), gliosis under the pia mater (two points).

**Table 1 jcm-09-01625-t001:** Our six cases of lepto/pachymeningitis secondary to RA activity.

Case	Gender (F/M)	Age of Diagnosis	Disease History	Neurological Symptoms	MRI	Lumbar Puncture	Treatment	Outcome (Follow-Up Duration)
**1**	F	57	0.5-year history of non-erosive RA, RF-, ACPA+, RA flare, csDMARD	Generalized seizure, headaches	Hypersignal in T2-weighted images and FLAIR mode Enhancement in T1-weighted images after IV gadolinium injection	Cell count: normal Proteins concentration: normal Glycorrhachia: normal	IV steroid pulses, IV Ig	Recovery with sequelae: ongoing anticonvulsant treatment, stable chronic MRI lesions (23 months)
**2**	M	69	7-years history of erosive RA, RF-, ACPA+, moderate RA activity, csDMARD + TNFα blocker	Headaches, frontal syndrome	Hypersignal in T2-weighted images and FLAIR mode Enhancement in T1-weighted images after IV gadolinium injection	Cell count: elevated Proteins concentration: normal Glycorrhachia: normal	IV steroid pulses, Etanercept, Rituximab	Death by subarachnoid hemorrhage (6 weeks after second Rituximab infusion)
**3**	F	63	25-years history of non-erosive RA, RF-, ACPA+, high RA activity, csDMARD	Generalize seizure, fluctuating fever, frontal syndrome, delirium, depression-anxiety syndrome	Hypersignal in T2-weighted images Enhancement in T1-weighted images after IV gadolinium injection	Cell count: elevated Proteins concentration: elevated Glycorrhachia: missing data Syphilis PCR: negative	IV steroid pulses, Etanercept, Adalimumab	Neurologic sequelae, stable chronic MRI lesions (24 months)
**4**	F	59	0.75-year history of non-erosive RA, RF+, ACPA+, RA remission, csDMARD + TNFα blocker	Headaches, psycho motor retardation	Enhancement in T1-weighted images after IV gadolinium injection	Cell count: elevated Proteins concentration: elevated Glycorrhachia: missing data *M. tuberculosis* PCR: negative	IV steroid pulses, Rituximab	Recovery with sequelae: ongoing anticonvulsant treatment, variable headaches, minor psycho motor retardation (11 months)
**5**	M	50	7-years history of erosive RA, RF+, ACPA+, moderate RA activity, csDMARD	Focal then generalized seizure, fever, alteration of the general state, dizziness with loss of consciousness, coma	Hypersignal in T2-weighted images Hyposignal in T1-weighted images without enhancement after injection	Cell count: elevated Proteins concentration: elevated Glycorrhachia: normal HIV PCR: negative	Only usual oral steroids	Recovery with sequelae: ongoing anticonvulsant treatment, stable chronic MRI lesions (10 years)
**6**	F	64	2-years history of non-erosive RA, RF-, ACPA+, low RA activity, no treatment	Seizure, fever, headaches, abnormal movements of the lower limb	Hypersignal in T2-weighted images and FLAIR mode Enhancement in T1-weighted images after IV gadolinium injection	Cell count: elevated Proteins concentration: normal Glycorrhachia: normal *M. tuberculosis* and Whipple PCR: negative	IV steroid pulses	Clinical and MRI recovery, stop anticonvulsant drugs (24 months)

M: male, F: female, RA: rheumatoid arthritis, RF: rheumatoid factors, ACPA: anti citrullinated peptide antibodies, HIV: human immunodeficiency virus, PCR: polymerase chain reaction, IV: intra venous, Ig: immunoglobulins, csDMARD: conventional synthetic disease modifying anti-rheumatic drug, TNF: tumor necrosis factor, MRI: magnetic resonance imaging

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
