# Peer review of "Rheumatoid Meningitis a Rare Extra-Articular Manifestation of Rheumatoid Arthritis: Report of 6 Cases and Literature Review"

_jcm, 2020, doi:10.3390/jcm9061625_

Round 1
Reviewer 1 Report
This is a nice report of 6 cases of RA presenting with meningitis. In addition the authors discuss the existing published literature on this rare association. My points are as follows:
- Disease activity of RA is presented as "low", "moderate" etc. Instead, we would like to see the actual DAS28 scores, or at least the number of affected TJC and SJC as more objective indices.
- Given the variable/inconsistent association with RA activity, coupled with the variable clinico-pathological features of meningitis in these patients, what is the proposed pathophysiology of this manifestations? Could this be discussed?
- Brain MRI. The lesions in brain parenchyma deserve some more detailed description. Did they point towards vasculitis? Were they UBOs due to the age and intake of glucocorticoids, or comorbidities etc.?
- Were brain MRI reviewed by a specific neurologist-neuroradiologist?
Author Response
Point 1-Disease activity of RA is presented as "low", "moderate" etc. Instead, we would like to see the actual DAS28 scores, or at least the number of affected TJC and SJC as more objective indices.
- Response 1 : TJC and SJC have been added in the « Results » section, page 4 lines 151-156. DAS28 was not exactly calculable for 3 patients, because of missing data (Patient Global Health or ESR/CRP), nevertheless we could classify all of them in different categories of disease activity (low/moderate/high).
Point 2-Given the variable/inconsistent association with RA activity, coupled with the variable clinico-pathological features of meningitis in these patients, what is the proposed pathophysiology of this manifestations? Could this be discussed?
- Response 2 : This is an interesting point, we thank the reviewer for raising it. We added in the “Discussion” section a paragraph, entitled “RM pathogenesis hypothesis” page 14 lines 347-368, and added some new references to discuss the pathophysiology of Rheumatoid Meningitis.
Point 3-Brain MRI. The lesions in brain parenchyma deserve some more detailed description. Did they point towards vasculitis? Were they UBOs due to the age and intake of glucocorticoids, or comorbidities etc.?
- Response 3 : No UBOs were described by neuroradiologist, and only one case were suspected of vaculitis on MRI but not confirmed. No intra parenchymal lesion was observed. Unfortunetaly, we don’t have more details about brain parenchyma lesions on MRI.
Point 4-Were brain MRI reviewed by a specific neurologist-neuroradiologist?
- Response 4 : MRI were reviewed by a neuroradiologist in at least 3 cases (Grenoble university hospital). Unfortunetaly, data were missing for 3 others. This is why we don’t point it out in the article.
Reviewer 2 Report
Dear author, the topic of this review is very interesting. Despite meningitis is rare, it represents a clinical challenge and a review could be very helpful. Some modifications are necessary:
- To describe in method section the range of time in which the six cases are registered and in how many centres.
- To insert in first table or in the text the cause of death of the case with bad outcome
- to describe in table 1 the infective investigation, main MRI findings, CSF findings.
- to organize the second table for clinical characteristic instead for every single case: for example described how many are female, the mean age, the mean disease duration, clinical onset either for neurological symptoms, either for RA, therapy.
- In conclusion to underline the possible clinical, diagnostic approach of patients with central nervous system manifestation.
Author Response
Point 1-To describe in method section the range of time in which the six cases are registered and in how many centres.
- Response 1 : These data have been added in the « Results » section, page 4 lines 136-138.
Point 2-To insert in first table or in the text the cause of death of the case with bad outcome.
- Response 2 : We agree with the reviewer that this point merits clarification. The cause of death for this patient was a subarachnoid hemorrhage. We added this information in Table 1.
Point 3-To describe in table 1 the infective investigation, main MRI findings, CSF findings.
- Response 3 : This is an interesting point. Lines have been added in Table 1 page 7 concerning MRI and lumbar puncture findings (please see the attachment). Biological investigations were not listed, because they were not homogeneous between the 6 patients and all did not performed exactly the same analysis. These data are presented in the « Results section » with more precision.
Point 4-To organize the second table for clinical characteristic instead for every single case: for example described how many are female, the mean age, the mean disease duration, clinical onset either for neurological symptoms, either for RA, therapy.
- Response 4 : A line with summarized data has been added in Table 2 page 11 (please see attachment). More precisions have been added in the « Discussion » section, pages 10 and 14.
5-In conclusion to underline the possible clinical, diagnostic approach of patients with central nervous system manifestation.
- The « Conclusion » section page 14 has been completed with a proposition of diagnosis approach according to the study and literature review.
